# Small Extracellular Vesicles and Oral Mucosa: The Power Couple in Regenerative Therapies?

**DOI:** 10.3390/cells13181514

**Published:** 2024-09-10

**Authors:** Blanka Maria Borowiec, Marta Dyszkiewicz-Konwińska, Dorota Bukowska, Michał Nowicki, Joanna Budna-Tukan

**Affiliations:** 1Department of Histology and Embryology, Poznan University of Medical Sciences, 60-781 Poznan, Polandmnowicki@ump.edu.pl (M.N.); 2Doctoral School, Poznan University of Medical Sciences, 60-812 Poznan, Poland; 3Department of Diagnostics, Poznan University of Medical Sciences, 60-812 Poznan, Poland; m.dyszkiewicz@ump.edu.pl; 4Department of Diagnostics and Clinical Sciences, Institute of Veterinary Medicine, Nicolaus Copernicus University in Torun, 87-100 Torun, Poland; dbukowska@umk.pl; 5Department of Anatomy and Histology, Collegium Medicum, University of Zielona Gora, 65-046 Zielona Gora, Poland

**Keywords:** small extracellular vesicles, sEVs, oral mucosa, regeneration, regenerative processes, wound healing, exosomes, review

## Abstract

Although ongoing debates persist over the scope of phenomena classified as regenerative processes, the most up-to-date definition of regeneration is the replacement or restoration of damaged or missing cells, tissues, organs, or body parts to full functionality. Despite extensive research on this topic, new methods in regenerative medicine are continually sought, and existing ones are being improved. Small extracellular vesicles (sEVs) have gained attention for their regenerative potential, as evidenced by existing studies conducted by independent research groups. Of particular interest are sEVs derived from the oral mucosa, a tissue renowned for its rapid regeneration and minimal scarring. While the individual regenerative potential of both sEVs and the oral mucosa is somewhat understood, the combined potential of sEVs derived from the oral mucosa has not been sufficiently explored and highlighted in the existing literature. Serving as a broad compendium, it aims to provide scientists with essential and detailed information on this subject, including the nature of the materials employed, isolation and analysis methodologies, and clinical applications. The content of this survey aims to facilitate the comparison of diverse methods for working with sEVs derived from the oral mucosa, aiding in the planning of research endeavors and identifying potential research gaps.

## 1. Introduction

While cells are the basic building blocks of all tissues, organs, and systems, it is known that many of the processes occurring in organisms are controlled by much smaller units. Regarded as highly prominent, exosomes are round-shaped, 30–150 nm-sized structures with lipid bilayer, secreted by cells, and involved in many biological cascades [1,2]. Currently, ongoing discussion persists regarding the nomenclature of extracellular vesicles. The primary challenge lies in distinguishing between different groups of extracellular vesicles. For example, both exosomes and microvesicles are types of extracellular vesicles, but they differ notably in their biogenesis [3]. Exosomes are formed through the exocytosis of multivesicular bodies [4], whereas microvesicles are produced by plasma membrane budding [5]. Size is another distinguishing factor, with exosomes previously classified as 30–100 nm and microvesicles ranging from 100–1000 nm [6]. However, this size overlap has complicated the identification of these vesicles, leading to considerable debate among scientists. To address this issue, the International Society for Extracellular Vesicles (ISEV) recommended a unified terminology in their 2018 document titled “Minimal Information for Studies of Extracellular Vesicles 2018 (MISEV2018): A Position Statement of the International Society for Extracellular Vesicles and Update of the MISEV2014 Guidelines.” [7]. It was an updated version of the earlier 2014 document (MISEV2014) [8]. The document from 2018 discouraged the use of the term “exosome”, unless the subcellular origin could be explicitly demonstrated. Instead, size-based terms such as “small extracellular vesicles” (<200 nm) and “medium/large extracellular vesicles” (>200 nm) were suggested, among others. The guidelines were further revised in 2023, but the <200 nm and >200 nm classification was not discouraged. However, the update emphasized that a strict consensus on size-based classification has not yet been reached, which is one of the reasons it advises increased caution when classifying vesicles based solely on size [9].

In our review, a significant portion of the studies still utilized the term “exosome” in their nomenclature. To contribute to the final unification of the nomenclature while being as close as possible to the authors’ definition, instead of the term “exosomes”, the term “small extracellular vesicles” (sEVs) will be used in accordance with the latest Minimal Information for Studies of Extracellular Vesicles (MISEV) recommendations [7,9].

Although research on sEVs began relatively recently, interest in them is growing steadily as new and promising functions and applications continue to be discovered. The first mentions of sEVs appeared over 40 years ago, in 1981 [10], yet they are still one of the most intensively studied structures involved in intercellular communication (Figure 1) Almost every cell type secretes small extracellular vesicles [1]. The functions of sEVs are numerous and vary depending on their origin and cargo (proteins, nucleic acids, lipids, or other cellular components) [11], shaped by the physiological or pathological condition of the cells [12]. According to the data from 2012, approximately 4400 proteins, 194 lipids, 1639 mRNAs, and 764 miRNAs have been identified in sEVs from various cell types, which illustrates their potential and functional complexity [13].

Small extracellular vesicles can be isolated from a wide variety of sources, both from humans and animals. These include, but are not limited to, stem cells [14,15,16], oral mucosa cells [17,18], epidermal cells [19,20], muscle cells [21,22], adipocytes [23], Wharton’s jelly mesenchymal cells [24,25], milk [26,27], urine [28,29], blood [30,31], semen [32,33], cerebrospinal fluid [34,35], amniotic fluid [36,37], and tears [38,39]. The relative ease of isolating sEVs, their availability, and the wealth of information they offer justify their examination from multiple perspectives, including the development of new therapies, particularly those focused on regeneration. The potential regenerative properties of sEVs have already been partially investigated using vesicles isolated from, among others, adipose tissue stem cells, neural stem cells, umbilical cord stem cells, plasma, and human exfoliated deciduous teeth [40,41,42,43,44].

The regenerative potential of sEVs from various sources has been recognized and continues to be studied. For example, the human adipose stem cells-derived sEVs were demonstrated to reduce high glucose-induced premature senescence of endothelial progenitor cells (EPCs) and improve wound healing in rats with diabetes. The same study showed that overexpression of nuclear factor erythroid 2-related factor 2 in these sEVs, further decreased EPCs senescence and improved wound healing by modulating protein expression. Since elevated glucose levels in diabetic patients activate the reactive oxygen species (ROS) and inflammation, which, in turn, cause EPC senescence and dysfunction, the use of adipose stem cells-sEVs to reduce EPC senescence could be a promising approach to treating diabetic foot ulcers [45,46]. A particularly interesting study by Tamura et al. investigated the immunosuppressive effects of sEVs derived from mesenchymal stem cells in a model of concanavalin A-induced liver injury [47]. In this mouse model, treatment with sEVs led to an increase in regulatory T cells, which are directly linked to the immune system. Additionally, there was an upregulation of TGF-β and hepatocyte growth factor mRNA levels, along with a reduction in hepatocyte apoptosis. The study also noted that sEVs derived from fibroblasts did not exhibit any immunoregulatory effects in this context. Interestingly, the effectiveness of sEVs was dose-dependent, with three injections proving to be optimal [47]. In addition to the studies mentioned above, there are numerous reports on sEVs from various sources influencing the regeneration of different cell types and tissues [45]. However, we believe that there is one particular tissue that has not received sufficient attention in this context.

Cells forming oral mucosa seem to be one of the most interesting sources of sEVs among those mentioned above. This mucous membrane lines the inside of the oral cavity, and it consists of three layers–stratified squamous epithelium, known as oral epithelium, connective tissue known as lamina propria and, in some cases, a dense connective tissue known as submucosa [48]. Although it varies in the degree of keratinization, the term “oral mucosa” covers areas of the cheeks, soft palate, alveolar mucosa, vestibular fornix and floor of the mouth (non-keratinized, movable mucosa), as well as the gingivae and hard palate (keratinized, masticatory mucosa). The surface of the tongue also has a specific type of mucosa that does not fit into the previous two categories, as it can be either keratinized or not, although some classifications consider it as masticatory mucosa. The placement of oral mucosa is an important factor while discussing its histology, as the third layer will be absent in locations where lamina propria is directly connected to the bone or muscle, as it is in gingiva, hard palate and some parts of tongue [49,50,51]. Literature indicates that the oral mucosa consists of four main groups of cells: epithelial cells, endothelial cells, fibroblasts, and immune cells [52]. The representation of the four main building blocks of barrier tissues is most similar between the oral mucosa and skin. However, the oral mucosa has a more extensive immune compartment, whereas the skin has a significantly larger epithelial compartment, by proportion [52]. Findings of Williams et al. delineate a complex array of epithelial and stromal cells in oral mucosal tissues, with diverse roles, especially in inflammation, antimicrobial defense, and neutrophil recruitment [52]. The availability of oral mucosa generally poses no issues, both for obtaining it as research material and for subsequent laboratory work. Besides commercial models such as SkinEthic™ Human Oral Epithelium (EPISKIN, Lyon, France) or EpiOral™ and EpiGingival™ tissues from MatTek Corporation (Ashland, MA, USA) [53], it can also be sourced from other origins with mandatory bioethical committee approval. For humans, clinical samples can be obtained from designated places like dental clinics [54] or hospitals [55]. For animals, similar collection procedures can be conducted in veterinary clinics [56], animal houses at universities or research institutes or slaughter houses [57]. Subsequent procedures largely depend on the type of research. The in vivo models, especially used for studying drug permeability, have been declining in popularity due to the increasing availability and preference for ex vivo animal models and in vitro cell culture 2D and 3D models [58]. Due to its ease of acquisition, oral mucosa is used in various types of research. These include, among others, drug permeability studies to evaluate drug penetration through the mucosal barrier [59], oral cancer research such as squamous cell carcinoma [60], immune response and inflammation studies [61], and tissue engineering for developing and testing new biomaterials [62]. Stem cells found in the oral mucosa are also used in therapies. However, these therapies may carry risks, such as tumor formation, genetic instability, immune rejection, and challenges in cell storage for administration [63,64]. It is worth noticing, that sEVs from stem and other types of cells, reduce some of these risks, such as possible acute immune response [65]. 

The accessibility and versatility of oral mucosa are significant reasons for its frequent use as a research model, but certainly there is more. Thanks to the atypical regenerative abilities of the oral mucosa, some scientists describe its healing processes as similar in some way to those found in [66]. This may be indicated by the phenotype of oral mucosal fibroblasts, which show similarities to foetal-type fibroblasts [67,68]. It is known that these fibroblasts contribute to foetal skin healing with minimal to no scarring [69]. This phenomenon combined with the fast regeneration of oral mucosa, highlights its potential in advancing wound healing and chronic wound treatment strategies not limited to the oral cavity [70,71]. The breakthrough also demonstrated the antibacterial properties of progenitor cells found in lamina propria of oral mucosa [72]. Possibly thanks to their constitutive secretion of osteoprotegerin (OPG) and haptoglobin (Hp), known for its antibacterial properties, the healing processes can proceed so efficiently [72].

Taken together, these findings imply that intercellular communication mediated by sEVs derived from oral mucosa might play a significant role in regenerative processes, though this will certainly require further exploration [45]. We would like to emphasize that there have been too few studies conducted on sEVs derived from the oral mucosa to definitively document their superiority over sEVs from other sources. We can only hypothesize that these vesicles produced by oral mucosal cells will play a significant role in accelerating regeneration processes. It therefore, seems reasonable to consider and explore their role in this process. The following review is a collection of reports on sEVs derived from the oral mucosa. It provides information on oral mucosa-derived sEVs management, including their isolation and subsequent analysis, as well as applications in biomedical sciences.

## 2. Technologies Used for Mucosa-Derived sEVs Processing

### 2.1. Isolation

There are various methods of isolating sEVs, and so far, no golden standard has been found to be ideal for a wide range of sEVs of different origins. While the help of the MISEV reports is definitely invaluable, due to the multiplicity of practice and the lack of unambiguous standardization, scientists still face problems with standardizing their results [73,74].

Published data show that differential centrifugation, also called ultracentrifugation, is the most frequently chosen method to isolate sEVs from the oral mucosa. Its main assumption is to modify the force and time of the samples’ centrifugation [75]. Individual particles sediment at different rates and those of greater density and size fall first. A series of subsequent centrifugations results each time in a pellet containing particles of lower sedimentation rate than the previous one [76]. The number of revolutions, speed and time of centrifugation vary slightly depending on the methodology developed by different research teams. For instance, according to the one described in 1996 by Raposo et al., which is still used today for isolating sEVs from cell culture media, ultracentrifugation can be categorized into two types: differential centrifugation and high-speed ultracentrifugation [77]. In differential centrifugation, the sample is centrifuged at progressively higher speeds, starting from 300–2000× *g* to remove large cells and debris, followed by a spin at 10,000× *g* to sediment larger extracellular vesicles and remaining debris. Finally, ultracentrifugation at 100,000× *g* is performed to isolate sEVs [77]. Lin et al. and Schwartz et al. used this method to isolate sEVs from culture media collected from oral mucosa cell cultures–human normal oral keratinocytes (NOKs) and OKF6/TERT2 cells (telomerase-immortalized human oral epithelial cell line) [78,79]. Similarly, Wang et al. applied this methodology for isolating sEVs from culture media from cultures of human oral mucosa epithelial cells [80]. In turn, to isolate sEVs from mesenchymal stem cells (MSCs) derived from oral mucosa tissues, Li et al. used a modification of differential centrifugation called density gradient differential centrifugation. In this method, isolation is based on size, mass, and density and requires the use of a previously constructed density gradient medium [81]. The density in such a medium decreases from the bottom to the top. Many types of gradient media are available nowadays; however, iodixanol and sucrose are the most commonly used for sEVs’ isolation [82]. With the use of this method, Kou et al. isolated sEVs from gingival mesenchymal stem cells (GMSCs) and skin mesenchymal stem cells (SMSCs) of mice [83] and Li et al. used this method for MSCs from human gingiva [84]. However, density gradient differential centrifugation is not always used for isolating sEVs from MSCs, as Wang et al., with other scientific teams, successfully isolated vesicles from GMSCs, using basic ultracentrifugation [85,86,87]. Reports of a method combining centrifugation and filtration when it comes to the isolation of sEVs are also reported. Coccè et al. used Microsep Advance Centrifugal Devices (Life Sciences, Port Washington, NY, USA) in their study, where, in addition to sample centrifugation, an element of ultrafiltration performed using special falcon-shaped 100 kDa filter devices is making an appearance [88]. Although these devices are used in different processes, such as concentrating microorganisms or clarifying samples, they are also used in the isolation of sEVs. According to the protocol, these devices are centrifuged at an acceleration of 3000 to 7500× *g*. Particles larger than the pores in the membrane placed in the device will remain in a special tank, while particles smaller in diameter will pass through the membrane and collect in the filtrate receiver. Using this method, sEVs derived from gingival mesenchymal stromal cells (GinPaMSCs) were successfully isolated [88].

Next in terms of frequency of use were special kits for the isolation of sEVs. The included set of reagents enables sEVs to be isolated in a short time and without the help of specialized equipment such as ultracentrifuges. Kits differ depending on their intended use and manufacturer, employing, among others techniques, such as size exclusion chromatography, immunological affinity, chemical precipitation, and simple centrifugation [89]. The exemplary protocol (here for ExoQuick-TC, System Biosciences, Palo Alto, CA, USA) alternates the simple steps of centrifugation, filtration, removal of the supernatant, and addition of the main reagent containing a proprietary polymer [90]. In turn to obtain sEVs from culture media of buccal mucosa MSCs transfected with human miR-185 mimic and human oral mucosa lamina propria-progenitor cells (OMLP-PCs) cultures, researchers successfully used GET™ Exosome Isolation Kit for stem cells (Genexosome Technologies Inc., Freehold, NJ, USA) and ExoSpin (Cell Guidance Systems, Cambridge, UK) [91]. A slightly different kit, Hieff™ Quick exosome isolation kit (YEASON, Shanghai, China), was used to isolate sEVs from GMSCs [92]. Due to the variety of produced reagents, consultations, and research, it is recommended that the most suitable kit for the research needs be selected. There are also reports of successful use of other kits, such as the MagCapture™ Exosome Isolation KIT PS (Fujifilm Wako Chemicals, U.S.A. Corp, Richmond, VA, USA), operating on affinity with the use of magnetic beads and phosphatidylserine (PS) binding protein. The use of PS binding protein ensures metal ion-dependent uptake of sEVs. Thanks to this, they are eluted from the magnetic beads, with the additional use of a metal chelating reagent with a neutral PH. The above kit was used for the isolation of sEVs from human GMSCs by Nakao et al. [87].

The final method was the size exclusion chromatography (SEC). Origins of this procedure date back to 1955, when Grant Lathe and Colin Ruthven used a column made of starch and water to perform the separation [93]. The technique is based on a porous stationary phase composed of spherical gel beads that sort the particles by size [94]. The components of the sample with a small hydrodynamic radius pass through the pores of the stationary phase, while particles with a large hydrodynamic radius (including sEVs) are not able to pass through the pores and can be eluted in subsequent steps [95,96]. In both experiments conducted by Sjöqvist et al., the qEV Izon columns (Izon Science, Addington, New Zealand) were used to obtain sEVs from culture media of oral mucosa cell cultures [17,18]. According to the manufacturer’s instructions, these columns separate particles based on size while passing through columns filled with porous polysaccharide resin. Smaller particles become trapped in the pores of the resin, which delays their passage through the column. The particles are collected in successive fractions from the largest to the smallest as they exit the column at different time intervals [97].

Although various methods for isolating sEVs exist, none of them is a true golden standard. Ultracentrifugation, both in its basic and modified forms, was the most commonly used method in the reviewed studies, appearing in 7 out of 14 papers. While modified versions did not report issues with sEV isolation, basic ultracentrifugation faced challenges, including incomplete separation of sEVs from microvesicles [78] and the presence of non-vesicular protein aggregates [79]. Sjöqvist et al. used size exclusion chromatography in their study, reporting lower sEV yields than expected, possibly due to degradation during cell culture [17]. Similar yield issues were noted with the Hieff™ Quick exosome isolation kit (YEASON, Shanghai, China) [92]. Even though many studies have not mentioned these challenges, it is crucial to consider both the literature and personal observations when planning experiments.

### 2.2. Analysis

Depending on the origin and destination of sEVs, their content, as well as their appearance and properties, change. Cargo provides valuable information on the role and purpose of these vesicles [12]. Among others, it can include nucleic acids, proteins, lipids and metabolites [98]. Methods of analysis are selected for each experiment, depending on specific needs and desired information.

Western blot (WB) is certainly the most popular method of analysis among the reviewed sources. This technique is widely used to analyze various proteins of interest, and in sEVs, cargo proteins are a perfect target. Briefly, WB uses the binding affinity of antibodies to target proteins (antigens). The reaction includes secondary antibodies often labeled with fluorescent dyes [3]. Although western blot is one of the most popular methods for characterizing sEVs, it is quite complex. Its procedure, described in more detail, primarily involves the successful isolation of sEVs, followed by their lysis, protein quantification using a protein assay, gel electrophoresis, subsequent transfer to a membrane, and blocking. Finally, it includes antibody incubation and detection using an imaging system [99]. The use of this technique is feasible because sEVs contain specific proteins either as their cargo or on their membrane, that can be recognized by antibodies [100]. The principles, as well as the advantages and disadvantages of this technique, have been extensively described previously [101,102]. Although very common, WB is a multi-step and complex analysis, carrying the treat of the final denaturation and reduction of proteins as well as the lysis of sEVs during the sample preparation [103], many scientists optimize the processes of carrying out the procedure and successfully use it to detect individual proteins carried within sEVs [17,18,78,82,83,84,85,91,103].

A technique that allows the quantification of sEVs and their size evaluation is named nanoparticle tracking analysis (NTA) (Figure 2). This procedure is able to determine the size of particles ranging from 10 to 1000 nm and therefore is ideally suited for the detection of sEVs. The Stokes-Einstein equation is used to calculate and estimate the size of the subjected particles. It is done by analyzing the laser light that scatters upon contact with the molecule [103]. The particle motion is analyzed by a camera, and then, thanks to the NTA software, the number and size of sEVs are estimated [104]. Although a relatively large sample volume (about 0.5 mL) is required, which can often be a challenge due to the very small number of cells from which sEVs are isolated, it is a vastly popular technique. An additional advantage of NTA is its capability for real-time sample analysis, as well as the use of fluorescent markers to identify specific populations of sEVs [105]. This popular method has also been widely studied and reviewed [105]. Sjöqvist et al., along with a few other research teams, used this technique in their oral mucosa-derived sEVs studies [17,78,84,85,86,87,90,91,103].

Electron microscopy is also a popular imaging technique for sEVs, and its variation- transmission electron microscopy (TEM) is particularly popular in the visualisation of sEVs from the oral mucosa. By using an electron beam in the microscope, high-resolution images of molecules with less than the size of a micron can be seen. The fluorescent screen in the microscope detects electrons that pass through the sample and do not interact with the molecules of interest [3] (for example sEVs), creating an image consisting of shadows and dark areas [106]. Similarly, though based on a slightly different principle than NTA, sEVs in TEM can also be labeled, allowing for the identification of specific groups. The precision of this technique also enables the examination of details such as the morphology of sEVs [107]. A disadvantage of this method is putative damage of sEVs in the drying process, which is required to prepare a valid sample for analysis [108]. Sjoqvist et al. [17,18], Yan Wang et al. [80] and also Schwartz et al. [79] decided to use TEM to analyze the isolated sEVs. Wenwen et al. were also using electron microscopy, however unspecified, to picture the sEVs [84]. The morphology of sEVs derived from GMSCs was also assessed by numerous research teams using TEM [82,84,85,86,87,88,90,91].

Although not all studies require it, in some cases, scientists choose sEV staining to follow their journey with an uptake assay. For this purpose, various dyes are used, and then the path or place of sEVs uptake in target cells is determined by means of visualization. Lin et al. isolated sEVs from the OKF6/TERT2 cell line and stained them with Vybrant dye (Invitrogen) to monitor their uptake in lymphoma cell line culture [78]. Another fluorescent lipophilic long-chain carbocyanine dye, PKH26 (Sigma-Aldrich, St. Louis, MO, USA), was used to stain sEVs isolated from buccal mucosa epithelial cells [17]. sEVs prepared this way were applied directly to the wound generated in the tissue fragment taken from the porcine esophagus. The wound bed was washed after incubation with sEVs and then observed under a fluorescent dissection microscope [17]. The same red fluorescent dye was used by Sjoqvist et al. to visualize the sEVs isolated from human oral keratinocytes (HOK) and normal human dermal fibroblasts (NHDF) [18]. The sEVs were applied to cell cultures of dermal fibroblasts and oral keratinocytes. Cells were then fixed in paraformaldehyde and observed under a confocal microscope [18]. In turn, for the uptake assay of GMSCs-derived sEVS, the ExoSparkler Exosome Membrane Labeling Kit-Green (Dojindo, Kumamoto, Japan) was successfully used by Nakao et al. [87]. 

Some scientists also isolate RNA and DNA from sEVs to analyze the nucleic acids trapped inside. It is known that some miRNAs derived from sEVs can bind and regulate mRNAs in recipient cells, while mRNAs derived from sEVs can be translated into particular proteins in receptor cells [95]. Much less was revealed about DNA carried by sEVs, but nevertheless, it is an excellent biomarker of pathological states of cells, and this property is successfully used in research [87]. Among many methods available for isolation of these nucleic acids from sEVs, Lin et al. used the miRNeasy minikit (Qiagen, Hilden, Germany) or TRIzol reagent (Life Technologies, Carlsbad, CA, USA) to isolate miRNA from oral mucosa vesicles [78]. The isolation was followed by an analysis with the use of NanoDrop 2000 (Thermo Fisher, Waltham, MA, USA) spectrophotometer and Agilent 2100 bioanalyzer with an RNA 6000 pico kit (Agilent, Santa Clara, CA, USA), performed to study the quantity and quality of RNA from said sEVs [78]. miR-200 family of miRNAs was the subject of interest in this research, and its quantification was evaluated using the TaqMan microRNA assay (Applied Biosystems, Waltham, MA, USA). Wang et al. were, in turn, separating total cellular RNA from oral epithelial-derived sEVs by 15% agarose gels to finally extract the small RNAs. Sequencing libraries were generated and then sequenced on an Illumina platform Hiseq 2500 (San Diego, CA, USA) by Annoroad Genomics (Beijing, China) [80]. In the study conducted by Schwartz et al., DNA was isolated from sEVs derived from human NOKs [79]. For this purpose, the XCF Exosomal DNA Isolation Kit (System Biosciences, Palo Alto, CA, USA) was used, and then the Nextera XT DNA kit (Illumina, San Diego, CA, USA) served for library preparation. Final sequencing was performed with NextSeq500 System (Ilumina, San Diego, CA, USA) with the use of the mid-output kit at paired-end 75 bp configuration [79]. The more extensive formation on isolating these cargoes has also been thoroughly described in the literature [109,110].

Among the reviewed publications, fluorescence-activated cell sorting (FACS), a derivative of the flow cytometry method, is also mentioned. Flow cytometry is excellent for the qualitative and quantitative analysis of particles, including sEVs, but their small size might be problematic [111]. This mainly applies to older instruments, as most of them have a detection limit of 300–500 nm, which largely exceeds the size of sEVs [112]. While analyzing sEVs by classic flow cytometry or FACS, it is necessary to immobilize them by, e.g., immunocapturing, on the surface of the beads used in this method to prevent their aggregation [3]. After incubation with labeled antibodies, the sample is subjected to acquisition, counted, and analyzed by the software [3]. Unlike classical flow cytometry, in FACS analysis, the particles are sorted and transferred to individual containers after passing through the flow cytometer [113]. Primarily, the use of FACS for analyzing sEVs is possible through the application of fluorescently labeled antibodies that bind to surface markers on sEVs, enabling their grouping based on these markers [114]. In the study conducted by Yang et al., sEVs isolated from OMLP-PCs were successfully analyzed with the use of this technique and the FACSCanto II Flow cytometer (BD Biosciences, Franklin Lakes, NJ, USA) equipped with a 488 nm and 535 nm laser excitation source [91].

When reviewing these studies, it’s important to note the extensive range of analyses conducted on sEVs. Given the many unknowns in this field, conducting as many analyses as possible is recommended to achieve a detailed understanding of the isolated vesicles. Such comprehensive insights could significantly enhance our understanding of this relatively young research area. The thoroughness of the teams’ work reviewed in this article has confirmed, among other findings, that through extensive analysis, Li et al. identified that miR-8485 is ectopically expressed on sEVs and has been confirmed to promote the proliferation and invasion of cancer cells. This suggests that targeting MSCs that secrete sEVs, or the sEVs themselves, could potentially delay or inhibit carcinogenesis [78]. Another example is the discovery that GMSCs secrete higher levels of interleukin-1 antagonist receptors compared to dermal MSCs, as revealed by Western Blot analysis. This finding may explain why wounds in gingival tissues heal better than those on the skin [84]. Through another set of analyses, including Western Blot, it was determined that sEVs are involved in gingival hyperplasia caused by idiopathic gingival fibroblasts. This suggests that targeting sEVs could help prevent excessive inflammation in the gingival area [91]. These findings underscore the importance of stringent guidelines for studying sEVs and other EVs, enabling a more precise understanding of their mechanisms. As a result, future research outcomes are expected to be increasingly accurate and insightful.

The technologies for processing mucosa-derived sEVs are summarized in Table 1.

## 3. Applications

Due to its diversity, universality, and ubiquity, sEVs have been receiving growing interest from the very beginning and have been generating a number of questions about their application in future biomedical sciences.

### 3.1. Selected Applications of sEVs of Various Origin

First of all, their potential is almost irreplaceable in terms of diagnostics [115]. One of the first times sEVs were used as a biomarker to diagnose a condition was in 1978 when these vesicles were isolated from cancer cells of a person with Hodgkin’s disease, and their level was found to be elevated in blood, urine, and serum of that patient [116,117,118]. Just 2 years later, in 1980, Poste and Nicolson made a pivotal discovery that brought a new dimension to sEVs perception [119]. Their study showed that even weakly metastatic B16 (F1) melanoma cells had become highly metastatic when fused with the sEVs isolated from the highly metastatic murine B16 (F10) melanoma cells. It suggested that F1 cells somehow acquire the feature of F10 cells via incorporating sEVs, thus opening the way for themselves to metastasize to the lungs [119]. sEVs are most widely used in cancer diagnostics, and they have also been applied in diagnostics of the central nervous system and cardiovascular diseases [120]. sEVs can also be used as targeted drug delivery systems. They were first used this way recently, in 2005, by Morse et al., who used immature dendritic cells from bone marrow as a source of sEVs treated as stimulatory molecules [117,121]. Regarding the regenerative potential of sEVs, they have already been used to study the repair of kidneys affected by acute inflammation [122] or parts of the brain affected by stroke [123]. They have also been shown to be able to regulate cell growth, promoting the development of rat neurites in in vitro cultured nerve cells [124]. Considering the above, the participation of sEVs in studying the process of wound healing, especially chronic wounds, as well as in the preparation of therapies enabling the treatment of these diseases, seems to be indispensable. sEVs are believed to play such an important role in the wound healing process, among others, due to their ability to initiate the epithelial reconstruction process by activating certain signaling pathways (AKT, ERK, STAT3, Wnt/β-catenin), leading to an increase in the expression of growth factors [125].

### 3.2. Documented Applications of Oral Mucosa-Derived sEVs

As noted previously, the origin of the sEVs used in therapy and diagnostics can vary, including oral mucosa. This tissue has a remarkable regenerative potential and is mostly deprived of the risk of scarring [126]. For example, sEVs isolated from oral mucosa epithelial cells (OMEC) sheets showed a pro-regenerative effect on skin wound healing, significantly reducing their size along the days of the study [17] (Figure 3). Scientists isolated sEVs from healthy human donors, among others with the aim of assessing the clinical potential of sEVs for epithelial stimulation and regeneration. They were isolated from conditioned and non-conditioned media. Human skin fibroblasts were treated with previously isolated sEVs. Depending on the dose, cells exposed to the vesicles showed a decrease in the proliferation of fibroblasts and a significant increase in the expression of growth factor genes such as *CTGF*, *HGF*, and *VEGFA*. To investigate the phenomenon of adhesion, sEVs were applied topically to wounds in the porcine esophagus ex vivo and then washed. The effect of sEVs administered to the cells could already be observed 1 min after the administration, but it increased with the time of co-incubation. The researchers also took it a step further and applied sEVs to wounds on the full-thickness skin of living rats. sEVs showed a visible effect as early as 5 days after administration, resulting in a significant reduction in the size of the wound. It is important to note that the reduced fibroblast proliferation observed in this study was anticipated. This reasoning is based on previous research demonstrating that local corticosteroid injections prevent the formation of esophageal strictures, which, like scars, are characterized by an abnormal increase of the production and deposition of collagen–protein majorly synthesized by fibroblasts [127,128]. The role of corticosteroids in preventing skin scarring by reducing fibroblast proliferation has also been independently confirmed, although it has been proven to be associated with an increased risk of oesophageal perforation [129]. A similar effect is observed with the use of mitomycin C, which, like corticosteroids, prevents esophageal strictures by inhibiting fibroblast proliferation [130]. Therefore, although decreased fibroblast proliferation might seem counterintuitive in the context of wound healing, it is indeed adequate. Additionally, Sjoqvist et al. suggest that their findings could contribute to the combined therapy of sEVs and cell sheets for future patients with early oesophageal cancer [17]. 

Other studies on sEVs derived from senescent normal oral keratinocytes (NOKs) have shown that they associate with mitochondrial and nuclear DNA, thereby activating interferon signaling in THP-1 monocytes in a STING-dependent manner [79]. This is well illustrated by the changes taking place in epithelial cells during aging and the role of sEVs in chronic inflammation and age-related diseases [79]. Schwartz also highlighted in his study that sEVs from senescent NOKs can activate IFN pathways in monocytes, which may adversely affect the normal and undisturbed course of the regeneration process. Attention was also drawn to the association of sEVs with the DNA blebbing out of the nucleus during aging, thus activating the cGAS-STING signaling responsible for inflammatory processes. This study raised question marks in many unobvious spots and, therefore, may be an insightful introduction to the issue of chronic inflammation related to premature cell aging and its possible inhibition or even reversal [79]. 

Continuing the topic of inflammation, there are reports that GMSC-derived sEVs influence the polarization and phenotype of macrophages in inflammatory conditions in the context of periodontitis [85]. It is characterized by the formation of a periodontal pocket, loss of attachment, and resorption of alveolar bone [131]. Considering that macrophages play a critical defensive role against pathogens causing periodontitis and can be polarized into pro-inflammatory (M1) and anti-inflammatory (M2) macrophages, this study showed that sEVs derived from GMSCs could inhibit the activation of M1 macrophages stimulated by lipopolysaccharides and interferon-γ while inducing them to convert to M2 macrophages. This finding may light new ways to treat macrophage-dominated periodontitis in humans [85]. The effect of GMSCs-derived sEVs on promoting the polarization of pro-inflammatory macrophages into an anti-inflammatory macrophage phenotype was also confirmed by Zhang et al. [132]. Returning to the periodontitis discussed previously, Nako et al. linked this disease to research on sEVs and macrophages [87]. Their study showed that preconditioning of GMSCs with tumor necrosis factor-alpha (TNF-α) increased CD73 gene expression in sEVs, thereby inducing anti-inflammatory polarization of M2 macrophages. When sEVS were injected locally into the palatal gingiva of mice, periodontal bone resorption decreased, as did the number of osteoclasts. sEVs derived from TNF-α conditioned GMSCs in this study demonstrated not only the ability to regulate inflammation but also to inhibit osteoclastogenesis [87]. Another inflammatory disease of the joints is collagen-induced arthritis (CIA). There are reports of CIA treatment with cell therapies using GMSCs, but they may carry risks such as tumorigenicity or immunogenicity [86]. The fact that sEVs carry a much lower risk of the above-mentioned complications was used by the team of Tian et al., who compared the effect of administering cells and sEVs separately to mice with CIA. The results of their study showed not only that sEVs from GMSCs have the same or stronger effect than GMSCs on stopping pro-inflammatory cytokine IL-17A and reducing the incidence and erosion of bones by inhibiting the IL-17RA-Act1-TRAF6-NF-kB signaling pathway [86]. It may, therefore, be concluded that sEVs therapies, which carry a lower risk of complications, have the potential to replace currently used cell therapies.

sEVs can also help deliver drugs to specific sites, as mentioned above. The paclitaxel (PTX) delivery pathway was investigated on sEVs derived from gingival mesenchymal stromal cells (GinPaMSC) [88]. This chemotherapy drug is used to treat some types of cancers, including breast, ovarian, and non-small cell lung cancer [133]. GinPaMSCs took up and then released paclitaxel in amounts that were pharmacologically effective against cancer cells. In the study conducted by Coccè et al., special attention was paid to sEVs, which turned out to incorporate PTX molecules during their biogenesis [88]. This discovery not only allows for a better understanding of the PTX delivery mechanism but also indicates that sEVs therapies may be an effective next step in the fight against PTX-responsive cancers.

Li et al. conducted a study in which mesenchymal stem cells (MSCs) derived from normal oral mucosa (N-MSC), oral cancer (Ca-MSC), and oral leukoplakia with dysplasia (LK-MSC) were isolated, and then put under investigation. His research team found that sEVs secreted by LK-MSC and Ca-MSC play an important role in promoting cell proliferation, migration, and invasion in vitro [84]. Interestingly, when sEVs secretion was blocked, the LK-MSC promoting effect was inhibited. During analysis of microarrays, scientists also identified sEVs-derived miR-8485, which was expressed ectopically and was able to promote the migration, proliferation, and invasion of neoplastic cells. These studies strongly suggest that interfering with the secretion of sEVs derived from MSCs, and thus stopping it, may prove to be an innovative strategy for delaying carcinogenesis [84]. The issue of both proliferation and apoptosis was also raised by Yin et al. [90]. The researchers investigated the effect of idiopathic gingival fibroblasts-derived sEVs, on the proliferation and apoptosis of normal gingival fibroblasts (N-GF). It was found that N-GFs co-cultured with sEVs showed an increase in Bcl-2, PCNA, and Ki67 levels and, in turn, a decrease in Bax levels. All studied genes were linked to either proliferation or apoptotic processes. The analyses undertaken in this study suggest, among other things, that sEVs derived from idiopathic gingival fibroblasts contribute to the regulation of proliferation and apoptosis of normal gingival fibroblasts [90]. These conclusions open new questions about a similar mechanism in fibroblasts of other origins. Although it is known that mesenchymal stem cells and sEVs play a special role in the wound healing process, the mechanism of exocytotic fusion leading to the secretion of sEVs and cytokines by these cells is not entirely clear. Kou et al. investigated this problem by simultaneously studying MSCs obtained from the gingivae and skin of mice [83]. The study showed that Fas receptor, which triggers a signal transduction pathway leading to apoptosis, in conjunction with Fas-associated phosphatase-1 (Fap-1) and caveolin-1 (Cav-1), activates a particular mechanism for the release of small extracellular vesicles (sEVs) in MSCs. It was also noted that MSCs secrete interleukin-1 receptor antagonist (IL-1RA) associated with sEVs, facilitating rapid gingival wound healing through the Fas/Fap-1/Cav-1 cascade. Although many conclusions can be drawn after analyzing this comprehensive study, it can be said that it unveils a previously undiscovered Fas/Fap-1/Cav-1 axis that governs SNARE-mediated sEV and IL-1RA secretion in stem cells, contributing to enhanced wound healing [83]. Even though this study provided a vast amount of new information, in order to be able to apply and implement the conclusions drawn from it in medicine, it would be necessary to study the same mechanisms in a human model.

However, it seems like sEVs isolated from healthy cells, such as oral keratinocytes and skin fibroblasts, inhibit the proliferation of those healthy cells, and it strongly depends on the dose of sEVs [18]. Studying oesophageal strictures, which are a common complication of oesophageal surgeries resulting in a reduction in the diameter of the oesophagus [134], Sjoqvist et al. isolated sEVs from conditioned media harvested from cultures of oral keratinocytes (“OKEx”) and skin fibroblasts (“FEx”). When studying the modulation of cell proliferation by sEVs, both groups were shown to inhibit the proliferation of healthy keratinocytes and fibroblasts. At some doses, sEVs acted similarly to the drug commonly used to prevent stricture, dexamethasone. In the case of non-tumorigenic HaCaT cells (spontaneously immortalized epithelial cells), a higher concentration of sEVs resulted in stronger suppression of proliferation, with dexamethasone showing similar suppression response. However, this pattern of activity was not replicated in cancer cells. Interestingly, these sEVs also inhibited the proliferation of the TR146 cancer cell line (it mimics normal human buccal epithelium), but dexamethasone had no reported effect there [18]. These reports may constitute the basis for undertaking research on the participation of sEVs in the therapy of stricture prevention [18].

Research on the regeneration of individual parts of the nervous system may also pin its hopes on sEVs therapies. According to reports by Rao et al., currently used methods of repairing peripheral nerve damage do not often give the expected results, as regeneration of this tissue is very slow [92]. The current preference for autologous nerve transplantation as a treatment for this condition is limited, primarily due to the need to sacrifice scarce healthy donor nerves and the potential for neuroma formation [135]. Previous studies have explored the use of GMSC transplantation for nerve regeneration [136,137], but the application of sEVs from GMSCs was not documented until Rao et al. investigated the effects of sEVs derived from GMSCs, in combination with biodegradable chitin wires, on peripheral nerve regeneration. It was decided to co-culture sEVs with Schwann cells and dorsal root ganglion axons using a rat model to regenerate the sciatic nerve defect. Due to the conducted analyses, it can be concluded that sEVs collected from GMSCs can promote Schwann cell proliferation and axon growth to a significant extent. Additionally, in vivo studies provided insight into the effect of sEVS on increasing the diameter and number of nerve fibers and the formation of myelin. Restoration of nerve conduction and muscle motor functions were also observed [92]. These promising results provide hope for the implementation of new peripheral nerve repair therapies using sEVs derived from GMSCs.

The connection between oral mucosa-derived sEVs and some viruses is also of great interest. Focusing on cancer, the human Epstein-Barr virus (EBV) should be underlined. EBV, which is transmitted from infiltrating B cells to epithelial cells, is causally related to certain types of cancer and lymphoma [138]. Li et al., while culturing oral epithelial cells (OKF6/TERT2), isolated sEVs containing high levels of miRNAs belonging to the miR-200 family. This family is specific for epithelial cells and promotes EBV lytic replication. Culturing OKF6/TERT2 cells with EBV-positive B cells induced viral reactivation. Additionally, the treatment of EBV-positive sEVs B cells derived from OKF6/TERT2 cells was a factor promoting viral reactivation. By deliberately using a cellular system that does not naturally express molecules of the miR-200 family, Li et al. found that forcing this expression is responsible for the generation of sEVs capable of inducing a lytic cascade in EBV-infected B cells. These studies suggest that the role of a tissue-specific environmental signal that initiates B cell reactivation, promoting viral transfer from peripheral B cell stores to the oral epithelium to facilitate amplification and virus exchange with other hosts, may be played by oral mucosa-derived sEVs [78]. This could mean that interfering with their function and structure could prevent the virus from spreading throughout the nearby tissues. A similar situation applies to enterovirus 71 (EV71). It causes, among others, hand, foot, and mouth disease (HFMD) in children [139]. By enzymatic digestion, Wang et al. separated human oral epithelial (OE) cells from the buccal mucosa in their study [80]. Small RNA deep sequencing was performed using cells infected with EV using CRISPR/Cas9, and the macrophage genome used in the study was altered by knocking out the myeloid differentiation factor 88 (MyD88) gene. The interaction of innate immune signaling networks with sEVs-derived miR-30a, which the literature claims may play a role in the excessive inflammatory response to viral infections [140], was also investigated. The study has shown that sEVs released from oral mucosal cells infected with EV71 are enriched in miR-30a [80]. Transfer of this particular miRNA into macrophages suppressed the type I interferon response by targeting MyD88 and facilitated viral replication in cells [80]. Given the above information, a hypothesis can be made about the possibility of regulating innate immunity against individual viruses by means of sEVs.

### 3.3. Possible Challenges Associated with Using Oral Mucosa-Derived sEVs for Regenerative Purposes

The limited number of clinical studies can likely be attributed to the challenges associated with working with sEVs. As in any field, the use of sEVs for regenerative purposes faces certain challenges. One major issue, as mentioned at the beginning of this article, is the heterogeneity of extracellular vesicles, including small extracellular vesicles (sEVs). Due to overlapping size ranges, extensive analyses are required to confirm not only the size but also the origin and content of the vesicles to ensure accurate isolation. The nomenclature and guidelines for these vesicles will likely continue to evolve and standardize over time. However, it is important to approach the classification of sEVs and other extracellular vesicles with caution [141].

Another challenge is the quantity of sEVs needed, as current applications require large amounts of vesicles for each administration or use. While cancer cells produce substantial quantities of sEVs, non-cancerous cells are significantly limited in this regard, and their production decreases further as they age. There are known methods to stimulate cells to increase sEV production by up to tenfold, but excessive stimulation may be harmful to the cells and degrade the quality of the vesicles [142]. It should be noted that, based on current knowledge, these vesicles must be derived from the host’s own cells, as they will be reintroduced into the same individual. To fully harness the regenerative potential of sEVs on a larger scale, it would be necessary to develop methods for widespread and rapid production of homogeneous sEVs that meet all required standards and regulations—a complex task given the instability of these structures. Regulations would not only apply to the production of sEVs but also, crucially, to their clinical implementation. As research on sEVs progresses, the number of regulations is expected to increase. While this is necessary, it will be essential to consider possible regulatory changes during ongoing research [143].

On a more fundamental and routine level, issues related to sEV research itself, rather than their direct implementation in regenerative medicine, include potential challenges with vesicle isolation. Although isolating these vesicles is relatively straightforward, not every isolation method is suitable for sEVs from every source. Therefore, it is crucial to allocate significant time to selecting and standardizing the appropriate method for the specific material being used [143].

Despite the numerous challenges, both potential and confirmed, they should not be seen as discouraging but rather as guiding new directions and pathways for research and development.

## 4. Conclusions

Regeneration is an indispensable part of every organism’s physiology. Although the field of regenerative medicine is rapidly advancing, the development of innovative regenerative therapies and the improvement of existing ones are crucial for progress in high-level medical care. Due to its broad scope, scientists are continually striving to better understand regenerative processes in organisms. A combination of two elements can be unusually helpful here—the oral mucosa, known for its mostly non-scarring and highly regenerative properties, and sEVs, known for their multitasking role in intercellular communication, strongly influencing tissue regeneration. 

Despite the wide range of the above-mentioned successful applications, clinical trials involving oral mucosa-derived sEVs are yet to come. For now, the ClinicalTrials.gov database shows no completed or ongoing clinical trials conducted with the use of oral mucosa-derived sEVs. This finding reveals a specific research gap, opening a window for the development of new innovative therapies or the improvement of existing ones. 

The authors believe that the methods of oral mucosa sEVs assessment that are profoundly described above carry great therapeutic potential and will trend over time. Therefore, not only for the purpose of this article but also with a perspective to use this name commonly, the term “ExOM therapies”, covering all therapies using sEVs originating from the oral mucosa, is proposed. We believe that this type of treatment will evolve with time to such an extent that separate nomenclature will be beneficial.

Despite being limited in number, the basic and clinical studies conducted to date and summarized in this review indicate unprecedented regenerative effects when using both elements simultaneously, underscoring the significant benefits of ExOM-based therapies. As in other well-known power couples, both components are strong and invaluable individuals, but only when combined can they create something truly peerless.

## Figures and Tables

**Figure 1 cells-13-01514-f001:**
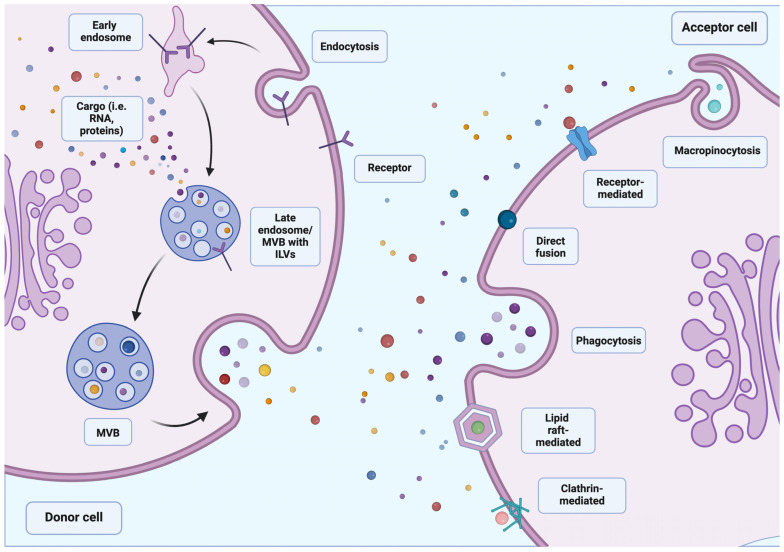
The diagram of cell communication pathways involving sEVS. The simplified production of sEVs presented above is initiated by the process of endocytosis. The early endosome formed as a result of this process becomes a late endosome, penetrating deep into the cell. As a result of the invagination of the lumen of the late endosome, intraluminal vesicles (ILVs) are formed, which absorb cargo (including nucleic acids, proteins, and peptides) at the moment of formation. At this point, the late endosome from the ILVs begins to move towards the cell membrane, simultaneously gaining the name multivesicular body (MVB). The MVB membrane then fuses with the cell membrane, releasing vesicles outside the cell, thus becoming sEVs. Depending on the type of their cargo, sEVs are divided and delegated to individual destinations. The part of vesicles that are directed to the cell can transfer their cargo to it via various pathways. These include direct fusion, macropinocytosis, phagocytosis, and connection mediated by receptors, lipid rafts, or clathrin. Created with BioRender.com.

**Figure 2 cells-13-01514-f002:**
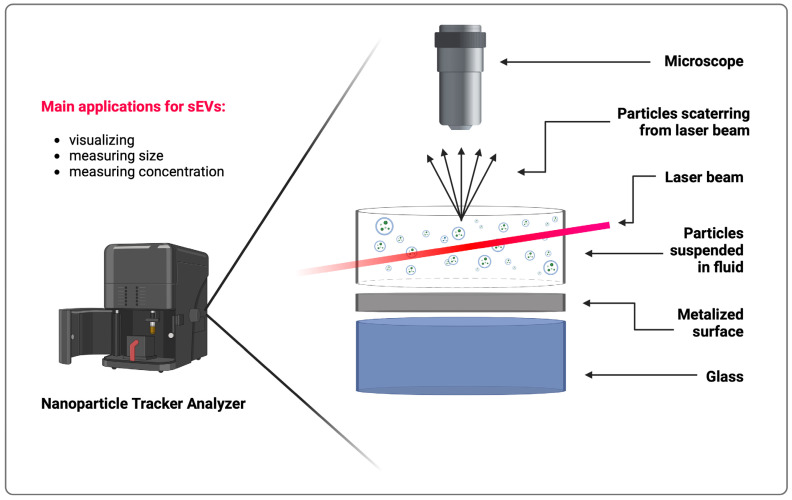
The principles of operating Nanoparticle Tracking Analyzer. Created with BioRender.com.

**Figure 3 cells-13-01514-f003:**
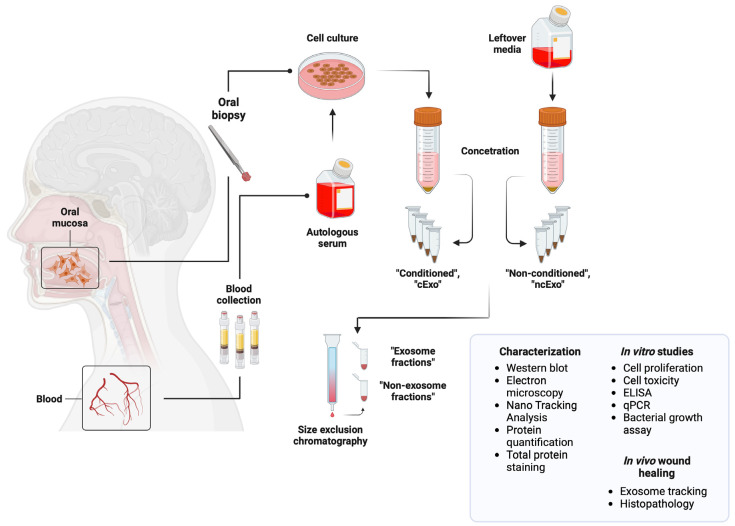
Visualization of the Sjoqvist study workflow [17], in which sEVs isolated from epithelial cell sheets of the oral mucosa showed a pro-regenerative effect on the healing of skin wounds, significantly reducing their size. Created with BioRender.com.

**Table 1 cells-13-01514-t001:** Summary of technologies for mucosa-derived sEVs processing.

Source of sEVs (Culture Media)	Type of Isolation	Main Types of Analysis	References
Gingival, bone marrow and skin MSCs	Density gradient differential centrifugation	TEM Western blot	[83]
Clinical-grade oral mucosal epithelial cell sheets	Size exclusion chromatography	TEM Uptake assay Western blot	[17]
Gingival MSCs	Ultracentrifugation	NTA TEM	[85,86,87]
		Uptake assay	[87]
		Western blot	[85,86]
	Ultracentrifugation with additional ultrafiltration	NTA TEM	[88]
	Kit for isolation	TEM Western blot	[92]
Idiopathic gingival fibroma fibroblasts	Kit for isolation	TEM	[90]
Normal oral keratinocytes	Ultracentrifugation	Isolating RNA or/and DNA NTA Uptake assay Western blot	[78]
OKF6/TERT2 cells	Ultracentrifugation	Isolating RNA or/and DNA TEM	[79]
Oral keratinocytes and dermal fibroblasts	Size exclusion chromatography	TEM Uptake assay Western blot	[18]
Oral mucosa epithelial cells	Ultracentrifugation	Isolating RNA or/and DNA NTA TEM Western blot	[80]
Oral mucosa lamina propria-progenitor cells	Kit for isolation	FACS NTA	[91]
Oral mucosal MSCs	Density gradient differential centrifugation	TEM	[84]

## Data Availability

Data sharing is not applicable to this article.

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
