# Peer review of "Small Extracellular Vesicles and Oral Mucosa: The Power Couple in Regenerative Therapies?"

_cells, 2024, doi:10.3390/cells13181514_

Round 1
Reviewer 1 Report
Comments and Suggestions for Authors
The manuscript by Borowiec et al. focus on the peculiarities of sEVs released by oral mucosa, focusing the attention on the evidence that oral mucosa has remarkable regenerative properties and so sEVs released from oral mucosa, which might represent an interesting player in tissue regeneration. The manuscript covers some interesting, although preliminary, studies. However, several issues should be addressed. Sections, such as 2.1 and 2.2, lack description of critical issues associated with different isolation methods or different characterization techniques. Discussion on critical issue associated with the use of sEVs for regenerative purposes are also lacking, so the possible obstacles that have led to the absence of Clinical Trials are not illustrated to readers.
Major points
In the Abstract, the sentence “renowned for rapid regeneration and minimal scarring observed in this tissue” is unclear, do authors refer to sEVs or to oral mucosa? The same is for the sentence “While individual potentials of sEVs and the oral mucosa are partially understood…”
In the introduction, it is impossible for readers that are not familiar with the topic to understand the reason underlying “…the use of the term “small extracellular vesicles” (sEVs) in place of the previously used and highly non-standardized term “exosomes””
The sentence “This phenomenon of minimal to no scarring highlights the potential of mucosal research in advancing wound healing and chronic wound treatment strategies.” Do authors mean in regions other that the oral mucosa? It is unclear
The concept of “..antibacterial properties of cells found in the mucosa” is unusual. What do authors mean? Intrinsic antibacterial properties outside the immune system cells? Could authors explain better?
At line 89, authors wrote that “All taken together suggest that intercellular communication mediated by sEVs plays an important role in mucosal regeneration [47]” However, the Introduction does not provide this evidence, as it includes a definition of sEVs and an introduction on the histology of oral mucosa
Lines 124-133 The introduction on ultracentrifugation is misleading, as it illustrates what happens by serial centrifugation of cell extracts. However, sEVs are isolated from culture medium. Authors should better explain this point
Line 144: In the sentence “With the use of this method Kou et al. isolated sEVs from gingiva and skin of mice [58]” it is not possible to understand if sEVs were isolated from tissue or from tissue-derived cell lines
The analysis section should provide more useful information: as for example, it is impossible to understand the reasons underlying the use of immunoblotting if authors do not explain that sEVs are characterized by the presence of specific protein markers. In addition, the methodological details of immunodetection should be familiar to the readers of Cells, the explanation is redundant.
In the case of section 2.2, a summary of the main findings on the physical-chemical properties of sEVs released from oral mucosa cells should be provided, otherwise it is just a list of studies without critical discussion
In section 3.2 the sentence “Depending on the dose, cells exposed to the vesicles showed a decrease in proliferation and a significant increase in the expression of growth factor genes such as CTGF, HGF and VEGFA.” seems contradictory: do sEVs induce regeneration by decreasing proliferation?
Line 351: The description of the study by Zhuang does not clarify why the evidence that “sEVs released by human gingival fibroblasts during radiotherapy may play a significant role in the osteogenic differentiation of hBMSCs” in terms of inhibition should open up new methods of treatments
Lines 471-485 The paragraph on nerve appears out of the scope of the review
Line 515: the paragraph on Clinical Trials is perhaps more appropriate in the Conclusions
The Conclusions paragraph lacks discussion on critical issues related to the applications of sEVs for regenerative purposes
Minor points
Please check the English of the sentence “…the interest in them grows proportionally to their constantly being revealed new and promising functions and applications”
The two paragraphs “Almost every cell type secretes small extracellular vesicles, including adipocytes, lymphocytes, platelets, muscle cells, stem cells, epithelial cells, nerve cells and oral mucosa cells [1,6–9].” and “These include, but are not limited to, stem cells [13–15], oral mucosa cells [7,16], epidermal cells [17,18], muscle cells [19,20], adipocytes [21], Wharton's jelly mesenchymal cells [22,23], milk [24,25], urine [26,27], blood [28,29], semen [30,31], cerebrospinal fluid [32,33], amniotic fluid [34,35] and tears [36,37]” are repetitive.”
Lines 354 “the MiR23a/CLCL12” should be “the miR23a/CLCL12”
Comments on the Quality of English LanguageThe Quality of English can be improved but it is generally understandable
Author Response
Reviewer 1
Comments and Suggestions for Authors
The manuscript by Borowiec et al. focus on the peculiarities of sEVs released by oral mucosa, focusing the attention on the evidence that oral mucosa has remarkable regenerative properties and so sEVs released from oral mucosa, which might represent an interesting player in tissue regeneration. The manuscript covers some interesting, although preliminary, studies. However, several issues should be addressed. Sections, such as 2.1 and 2.2, lack description of critical issues associated with different isolation methods or different characterization techniques. Discussion on critical issue associated with the use of sEVs for regenerative purposes are also lacking, so the possible obstacles that have led to the absence of Clinical Trials are not illustrated to readers.
Thank you for taking the time to review our manuscript. We greatly appreciate this valuable feedback and insights, which will undoubtedly help improve the quality of the work.
Major points
- In the Abstract, the sentence “renowned for rapid regeneration and minimal scarring observed in this tissue” is unclear, do authors refer to sEVs or to oral mucosa? The same is for the sentence “While individual potentials of sEVs and the oral mucosa are partially understood…”
The first sentence is now changed to ”Of particular interest are sEVs derived from the oral mucosa, the tissue renowned for rapid regeneration and minimal scarring observed in this tissue.”
In the second sentence we aimed to emphasize, that while the individual regenerative potential of sEVs (separately) and the regenerative potential of oral mucosa (separately) is somewhat understood, the combined potential of sEVs derived from the oral mucosa has not been sufficiently explored or highlighted in the existing literature. This suggests that more attention should be given to how these components might work together to enhance regenerative processes or other biological functions.
To clarify it, the sentence is now changed to “While individual regenerative potential of both sEVs and the oral mucosa is somewhat understood, the combined potential of sEVs derived from the oral mucosa has not been sufficiently explored and highlighted in the existing literature.”
- In the introduction, it is impossible for readers that are not familiar with the topic to understand the reason underlying “…the use of the term “small extracellular vesicles” (sEVs) in place of the previously used and highly non-standardized term “exosomes””
To clarify this issue, we added a paragraph in Introduction section.
- The sentence “This phenomenon of minimal to no scarring highlights the potential of mucosal research in advancing wound healing and chronic wound treatment strategies.” Do authors mean in regions other that the oral mucosa? It is unclear
The sentence is now changed to “This phenomenon combined with fast regeneration of oral mucosa highlights its potential in advancing wound healing and chronic wound treatment strategies, that are not limited to oral cavity.”
- The concept of “..antibacterial properties of cells found in the mucosa” is unusual. What do authors mean? Intrinsic antibacterial properties outside the immune system cells? Could authors explain better?
The sentence is now changed to “The breakthrough was also the demonstration of the antibacterial properties of pro-genitor cells found in lamina propria of oral mucosa [54]. Possibly thanks to their constitutive secretion of osteoprotegerin (OPG) and haptoglobin (Hp), known for its antibacterial properties, the healing processes can be proceeded so efficiently [55].”
- At line 89, authors wrote that “All taken together suggest that intercellular communication mediated by sEVs plays an important role in mucosal regeneration [47]” However, the Introduction does not provide this evidence, as it includes a definition of sEVs and an introduction on the histology of oral mucosa
Thank you for bringing this important issue to our notice. The whole new paragraph has been added to the section (125-132) and the above mentioned sentence has been slightly changed to “Taken together, these findings imply that intercellular communication mediated by sEVs derived from oral mucosa might play a significant role in regenerative processes, though this will certainly require further exploration.”
- Lines 124-133 The introduction on ultracentrifugation is misleading, as it illustrates what happens by serial centrifugation of cell extracts. However, sEVs are isolated from culture medium. Authors should better explain this point
Thank you for the opportunity to clarify. This fragment has been modified.
- Line 144: In the sentence “With the use of this method Kou et al. isolated sEVs from gingiva and skin of mice [58]” it is not possible to understand if sEVs were isolated from tissue or from tissue-derived cell lines
The sentence has been changed to “With the use of this method Kou et al. isolated sEVs from gingival mesenchymal stem cells (GMSCs) and skin mesenchymal stem cells (SMSCs) of mice.”.
- The analysis section should provide more useful information: as for example, it is impossible to understand the reasons underlying the use of immunoblotting if authors do not explain that sEVs are characterized by the presence of specific protein markers. In addition, the methodological details of immunodetection should be familiar to the readers of Cells, the explanation is redundant.
More details and information are now added.
- In the case of section 2.2, a summary of the main findings on the physical-chemical properties of sEVs released from oral mucosa cells should be provided, otherwise it is just a list of studies without critical discussion
The summaries for both sections have been added. Thank you for this valuable suggestion.
- In section 3.2 the sentence “Depending on the dose, cells exposed to the vesicles showed a decrease in proliferation and a significant increase in the expression of growth factor genes such as CTGF, HGF and VEGFA.” seems contradictory: do sEVs induce regeneration by decreasing proliferation?
Thank you for bringing this aspect to our attention. We clarified it, as the decrease of fibroblasts’ proliferation is consistent with the article. This reasoning is based on previous research demonstrating, that local corticosteroid injections prevent the formation of esophageal strictures, which, like scars, are characterized by abnormal increase of production and deposition of collagen – protein majorly synthesized by fibroblasts. The role of corticosteroids in preventing skin scarring by reducing fibroblast proliferation has also been independently confirmed. The following paragraph has been added: “. It is important to note that the reduced fibroblast proliferation observed in this study was anticipated. This reasoning is based on previous research demonstrating, that local corticosteroid injections prevent the formation of esophageal strictures, which, like scars, are characterized by abnormal increase of production and deposition of collagen – protein majorly synthesized by fibroblasts. The role of corticosteroids in preventing skin scarring by reducing fibroblast proliferation has also been independently confirmed. Therefore, although decreased fibroblast proliferation might seem counterintuitive in the context of wound healing, it is indeed adequate.”
- Line 351: The description of the study by Zhuang does not clarify why the evidence that “sEVs released by human gingival fibroblasts during radiotherapy may play a significant role in the osteogenic differentiation of hBMSCs” in terms of inhibition should open up new methods of treatments
Thank you for highlighting this matter, we apologize for this omission. This information has been deleted from the text and the table. The corrected table is already in the text, but it will be additionally uploaded to the system.
- Lines 471-485 The paragraph on nerve appears out of the scope of the review
While our review focuses on the oral mucosa, it does so primarily in the context of this tissue being a source of sEVs that can positively influence regenerative processes in various parts of the body, not just the oral mucosa. Therefore, in the "Applications" section, we do not limit the discussion to the effects of oral mucosa-derived sEVs on the oral cavity alone. While the majority of research on oral mucosa-derived sEVs is associated with regeneration in the oral cavity, their regenerative potential has also been observed in other areas, such as peripheral nerves or the esophagus. This aspects may play an important role in exploring new areas of the oral mucosa-dreived sEVs, therefore, we decided not to overlook these important publications in our review.
- Line 515: the paragraph on Clinical Trials is perhaps more appropriate in the Conclusions
We added this paragraph to the Conclusions section.
- The Conclusions paragraph lacks discussion on critical issues related to the applications of sEVs for regenerative purposes
Thank you for this suggestion. We added new paragraphs as advised, but we placed it in a separate section.
Minor points
- Please check the English of the sentence “…the interest in them grows proportionally to their constantly being revealed new and promising functions and applications”
The sentence has been changed to “Although research on sEVs began relatively recently, interest in them is growing steadily as new and promising functions and applications continue to be discovered.”
- The two paragraphs “Almost every cell type secretes small extracellular vesicles, including adipocytes, lymphocytes, platelets, muscle cells, stem cells, epithelial cells, nerve cells and oral mucosa cells [1,6–9].” and “These include, but are not limited to, stem cells [13–15], oral mucosa cells [7,16], epidermal cells [17,18], muscle cells [19,20], adipocytes [21], Wharton's jelly mesenchymal cells [22,23], milk [24,25], urine [26,27], blood [28,29], semen [30,31], cerebrospinal fluid [32,33], amniotic fluid [34,35] and tears [36,37]” are repetitive.”
The first paragraph is now very brief (Almost every cell type secretes small extracellular vesicles [1].), while the second paragraph remained the same.
- Lines 354 “the MiR23a/CLCL12” should be “the miR23a/CLCL12”
The paragraph with the sentence containing this phrase has been deleted (Comment number 11).

Reviewer 2 Report
Comments and Suggestions for Authors
This review covers sEVs derived from the oral mucosa. This is a relatively unique topic for a review, and is interesting, in part, due to the unusually high regenerative potential for this tissue. Similarly, sEVs have attracted signficant attention in recent years, in part due to their role in regenerative medicine. Overall, this is a novel and timely review.
This review is reasonably comprehensive given its focus on sEVs specific to the oral mucosa. The authors also cover isolation and analysis methods as well as potential clinical applications.
Figures and table are helpful and nicely done. References are extensive. While a somewhat esoteric topic, even researchers outside this specific field are likely to find this review interesting and helpful.
Author Response
Reviewer 2
This review covers sEVs derived from the oral mucosa. This is a relatively unique topic for a review, and is interesting, in part, due to the unusually high regenerative potential for this tissue. Similarly, sEVs have attracted signficant attention in recent years, in part due to their role in regenerative medicine. Overall, this is a novel and timely review.
This review is reasonably comprehensive given its focus on sEVs specific to the oral mucosa. The authors also cover isolation and analysis methods as well as potential clinical applications.
Figures and table are helpful and nicely done. References are extensive. While a somewhat esoteric topic, even researchers outside this specific field are likely to find this review interesting and helpful.
We greatly appreciate your positive feedback on our work. We believe it will serve as a valuable resource for researchers planning studies on sEVs from oral mucosa, as well as those reviewing literature for challenges encountered by other research teams. While this area is not yet extensively explored, we are confident that its potential will soon drive further investigation and advancements in the field.

Reviewer 3 Report
Comments and Suggestions for Authors
The submitted review article focuses on the utility of oral mucosal extracellular vesicles for regenerative purposes. Isolation and characterization are not different for the EVs isolated from other sources and, therefore, it is important to highlight the features that make these EVs better for regeneration than EVs isolated from other sources.
Comments
- It is essential to include information on tissue specificity, availability, applications, differences in the effect of EVs from different sources in regeneration, and risks associated with the use of non-fully differentiated cells in general.
-The authors state that they are limiting their review to sEVs, formerly known as exosomes. What is the reason for this?
-Figure 3 shows a workflow for sEV isolation and characterization, but the authors do not comment on which workflow they recommend.
Author Response
The submitted review article focuses on the utility of oral mucosal extracellular vesicles for regenerative purposes. Isolation and characterization are not different for the EVs isolated from other sources and, therefore, it is important to highlight the features that make these EVs better for regeneration than EVs isolated from other sources.
Thank you for taking the time to review our manuscript. We greatly appreciate this valuable feedback and insights, which will undoubtedly help improve the quality of the work.
Comments
- It is essential to include information on tissue specificity, availability, applications, differences in the effect of EVs from different sources in regeneration, and risks associated with the use of non-fully differentiated cells in general.
The information has been added to the Introduction section.
-The authors state that they are limiting their review to sEVs, formerly known as exosomes. What is the reason for this?
The majority of studies were conducted during the period when the nomenclature for extracellular vesicles (EVs) was being standardized. Almost all the studies included in this review, found according to the specified criteria, used the term "exosomes." While the classification of intercellular vesicles has since evolved, we aimed to accurately reflect the authors' original intent by updating the terminology from "exosomes" to "small extracellular vesicles (sEVs)." Since the standardization of nomenclature, ongoing research likely incorporates the new classification, possibly including newly identified types of EVs such as autophagic EVs, stress-induced EVs, and matrix vesicles.
-Figure 3 shows a workflow for sEV isolation and characterization, but the authors do not comment on which workflow they recommend.
Thank you for this comment. To create this figure, we chose a workflow that was complex and sometimes difficult to follow. Given that the aim of this review is to facilitate the use of already published studies by compiling them in one place, we believed it was appropriate to visually present the most complex workflow among the selected studies. Additionally, our work did not aim to evaluate or recommend specific studies. Its purpose was solely to compile information on sEV processing found in the cited works.

Round 2
Reviewer 1 Report
Comments and Suggestions for Authors
The manuscript by Borowiec et al. has addressed all raised issues. However, in their reply authors suggested that the regenerative potential of oral mucosa sEVs has also been observed in other areas, such as peripheral nerves or the esophagus. Their considerations are useful to be introduced also to readers.
Author Response
Reviewer 1
The manuscript by Borowiec et al. has addressed all raised issues. However, in their reply authors suggested that the regenerative potential of oral mucosa sEVs has also been observed in other areas, such as peripheral nerves or the esophagus. Their considerations are useful to be introduced also to readers.
Thank you for this comment. In the last version of the manuscript, the considerations regarding the peripheral nerve and oesophagus are addressed in lines 425-449, 534-545 and 546-560, respectively. However, we have now added additional information to improve understanding of these matters, leading to changes in lines 425-451, 536-551, and 552-569, respectively.

Reviewer 3 Report
Comments and Suggestions for Authors
The authors addressed my comments.
Author Response
Reviewer 3
The authors addressed my comments.
We greatly appreciate your feedback on our work, which has contributed to its improvement. We believe it will serve as a valuable resource for researchers planning studies on sEVs from oral mucosa, as well as those reviewing literature for challenges encountered by other research teams in the past. While this area is not yet extensively explored, we are confident that its potential will soon drive further investigation and advancements in the field.
